# Molecular detection and characterisation of the first Japanese encephalitis virus belonging to genotype IV acquired in Australia

Chisha Sikazwe[1☯], Matthew J. Neave[2☯], Alice Michie[1], Patrick Mileto[2], Jianning Wang[2], Natalie Cooper[1], Avram Levy[1,3], Allison Imrie[1,3], Robert W. Baird[4], Bart J. Currie[4], David Speers[1], John S. Mackenzie[1,5], David W. Smith[1]*, David T. Williams[2]*

**1** PathWest Laboratory Medicine Western Australia, Nedlands, Western Australia, Australia, **2** CSIRO Australian Centre for Disease Preparedness, Geelong, Victoria, Australia, **3** School of Biomedical Sciences, University of Western Australia, Nedlands, Western Australia, Australia, **4** Pathology and Infectious Diseases Departments, Royal Darwin Hospital, Darwin, Northern Territory, Australia, **5** Faculty of Health Sciences, Curtin University, Bentley, Western Australia, Australia

☯ These authors contributed equally to this work.
* david.smith@health.wa.gov.au (DWS); d.williams@csiro.au (DTW)

**Data Availability Statement:** The final whole genome sequence of JEV/Australia/NT_Tiwi Islands/2021 was deposited in the NCBI GenBank

## Abstract

### Background

A fatal case of Japanese encephalitis (JE) occurred in a resident of the Tiwi Islands, in the Northern Territory of Australia in February 2021, preceding the large JE outbreak in south-eastern Australia in 2022. This study reports the detection, whole genome sequencing and analysis of the virus responsible (designated JEV/Australia/NT_Tiwi Islands/2021).

### Methods

Reverse transcription quantitative PCR (RT-qPCR) testing was performed on post-mortem brain specimens using a range of JE virus (JEV)-specific assays. Virus isolation from brain specimens was attempted by inoculation of mosquito and mammalian cells or embryonated chicken eggs. Whole genome sequencing was undertaken using a combination of Illumina next generation sequencing methodologies, including a tiling amplicon approach. Phylogenetic and selection analyses were performed using alignments of the Tiwi Islands JEV genome and envelope (E) protein gene sequences and publicly available JEV sequences.

### Results

Virus isolation was unsuccessful and JEV RNA was detected only by RT-qPCR assays capable of detecting all JEV genotypes. Phylogenetic analysis revealed that the Tiwi Islands strain is a divergent member of genotype IV (GIV) and is closely related to the 2022 Australian outbreak virus (99.8% nucleotide identity). The Australian strains share highest levels of nucleotide identity with Indonesian viruses from 2017 and 2019 (96.7–96.8%). The most recent common ancestor of this Australian-Indonesian clade was estimated to have emerged in 2007 (95% HPD range: 1998–2014). Positive selection was detected using two

database (Accession no. OM867669). All relevant data are within the manuscript and its Supporting Information files.

**Funding:** The authors received no specific funding for this work.

**Competing interests:** The authors have declared that no competing interests exist.

methods (MEME and FEL) at several sites in the E and non-structural protein genes, including a single site in the E protein (S194N) unique to the Australian GIV strains.

## Conclusion

This case represents the first detection of GIV JEV acquired in Australia, and only the second confirmed fatal human infection with a GIV JEV strain. The close phylogenetic relationship between the Tiwi Islands strain and recent Indonesian viruses is indicative of the origin of this novel GIV lineage, which we estimate has circulated in the region for several years prior to the Tiwi Islands case.

### Author summary

A fatal human case of Japanese encephalitis (JE) occurred in a patient from the Tiwi Islands of northern Australia in February 2021. The Tiwi Islands are 80km north of Darwin in the Timor Sea. Attempts to culture the virus from post-mortem brain tissue were unsuccessful. However, the whole genome was successfully sequenced and compared phylogenetically to other JE viruses. The Tiwi Islands strain was shown to belong to the rarely detected genotype IV (GIV) of the JE virus (JEV), together with the Australian 2022 outbreak strain, and is only the second fatal case of JE associated with a GIV virus. JEV strains isolated from Indonesia in 2017 and 2019 were shown to be the most-closely-related to the Australian GIV strain providing evidence for the geographic origins of the emergent Australian virus. From evolutionary analysis, the clade containing the Australian and recent Indonesian viruses was estimated to have emerged between 1998 and 2014, suggesting that this lineage of GIV viruses has been circulating for several years before the Tiwi Islands case. This is the third JEV genotype to be detected in Australia and demonstrates the ease with which new genotypes can spread and unexpectedly cause disease in new areas. The possible origin and risks of further incursions into Australia are discussed.

## Introduction

Japanese encephalitis virus (JEV), a zoonotic mosquito-borne flavivirus, is the major viral cause of encephalitis in Asia, with approximately 68,000 cases typically occurring annually [1], but may exceed 100,000 cases in some years [2]. Most human infections are asymptomatic, with an estimated less than 1% of cases resulting in a clinical disease that ranges from a mild febrile illness to severe meningomyeloencephalitis. Approximately 25% of clinical cases are fatal, with a further 50% of cases resulting in permanent neuropsychiatric sequelae [3]. It is well established that the virus exists in a zoonotic transmission cycle between ardeid water birds, such as herons and egrets, and *Culex* mosquitoes, particularly *Culex tritaeniorhynchus*. Domestic pigs are the major amplifying host [4–6]. The virus occurs as five phylogenetically distinct genotypes (GI-GV), each with its own geographic distribution pattern [7]. All five genotypes are found in the Indo-Malaysia region, where JEV is endemic and believed to have originally emerged [8], but elsewhere in Asia and Oceania, between one and three specific genotypes have coexisted in different geographic areas over different time periods [7,9].

JEV has demonstrated a strong propensity to spread and establish in new areas [5,6,8,10,11]. In 1995 JEV emerged for the first time in the Australasian region when three cases of encephalitis, two of which were fatal, occurred on Badu, an island in the central Torres Strait (S1 Table) [12,13]. The virus was isolated from both subclinical human infections and from *Cx. annulirostris* mosquitoes [12,14]. Further incursions occurred in 1998 when two encephalitis cases were reported, one from Badu and the first case of JE on mainland Australia in south-western Cape York [15]. Serological evidence of JEV infection was also observed in pigs close to the site of the human case on Cape York. Partial gene sequence analysis of virus isolates obtained in 1995 and 1998 showed that these were nearly identical [15] and belonged to GII, and distantly related to JEV isolates from Indonesia [16].

Investigations into the origin of JEV in northern Australia strongly implicated Papua New Guinea (PNG) as the source; three isolates of JEV were obtained from *Cx. sitiens* subgroup mosquitoes trapped in Western Province [17]; serological evidence of infection from human and porcine sera collected from various sites in PNG between 1989 and 1997 [10,18]; and clinical cases of JE reported across the New Guinea island [10,19–21]. PNG mosquito isolates also showed high levels of nucleotide identity (>99%) with isolates obtained from Badu in 1995 and 1998 [17]. It has been suggested that the JEV-infected mosquitoes were blown into northern Australia from PNG by cyclonic winds [22]. A further incursion of JEV into the Torres Strait was observed in 2000, but no human case was reported [23]. Further virus isolates were obtained from *Cx. gelidus* mosquitoes and porcine sera on Badu, and were found to cluster in a different genotype, GI, and were most closely related to isolates from Thailand [24]. How this new genotype reached Australia remains unknown. Further isolates were obtained from mosquitoes trapped on Badu and the Northern Peninsula Area of Cape York in 2004 [25]. The latter represents the first isolate of JEV from mainland Australia. Serological surveillance activities by the Northern Australian Quarantine Strategy program showed frequent JEV activity in the following years in the Torres Strait (2005, 2010, 2012, 2013, 2014, 2016 and 2017) and Northern Peninsula Area (2019, 2020 and 2021; data extracted from Animal Health Surveillance Quarterly Reports, https://animalhealthaustralia.com.au/supporting-market-access/). A JE vaccination program continues in the outer Torres Strait Islands and no further human JE cases have been reported from that region since 1998.

The first locally acquired Australian detection of JEV outside the Torres Strait Islands or northern Queensland occurred in a 45-year-old person living on the Tiwi Islands, located 80km north of Darwin (Northern Territory) in the Timor Sea between Australia and Indonesia. The patient, who had not travelled outside their community, was hospitalised in February 2021 with a 2 day history of acute confusion and fever [26]. Flavivirus encephalitis was suspected based on MRI scan and serological results, but premortem PCR tests were negative. The patient had rapid and progressive neurological deterioration and died 15 days after admission to the intensive care unit of Royal Darwin Hospital. This case was a harbinger for the large, multi-state outbreak in south-eastern Australia in 2022 [27,28]. In this paper, we describe the detection, whole genome sequencing, phylogenetic and evolutionary analysis of the virus responsible for the fatal Tiwi Islands case, and consider the possible origins of the virus, and the risks of further incursions of new genotypes into northern Australia.

## Materials and methods

### Ethics statement

The use of embryonated chicken eggs (ECE) was conducted with the approval of the Commonwealth Scientific and Industrial Research Organisation (CSIRO) Australian Centre for Disease Preparedness Animal Ethics Committee (permit number AEC 2023). All procedures

were conducted according to the guidelines of the National Health and Medical Research Council as described in the Australian code for the care and use of animals for scientific purposes [29].

## Samples and virus detection

Postmortem brain tissue samples from the clinical case of JE in the Tiwi Islands were sent from Northern Territory Pathology to PathWest Laboratory Medicine in Western Australia (PWLM) and the ACDP, Victoria, for further laboratory investigation. The Tiwi Islands, comprising Melville Island and Bathurst Island, are part of the Northern Territory of Australia, located at 11˚36′S 130˚49′E (Fig 1). At ACDP, a 10% (w/v) homogenate of the right thalamus was prepared in Dulbecco's PBS (ph 7.6; Oxoid) containing antibiotics (400 IU/ml of penicillin and 400 μg/ml of streptomycin; Sigma-Aldrich) using a 3 ml syringe and 18G blunt needle. Samples were clarified by low-speed centrifugation (1000×g, 5 min, 4˚C) and the supernatant used for RNA extraction and virus isolation. At PWLM, homogenate was derived from a 1 cm$^3$ sample of post-mortem brain tissue (n = 5) in a mortar pre-chilled at –20˚C and ground

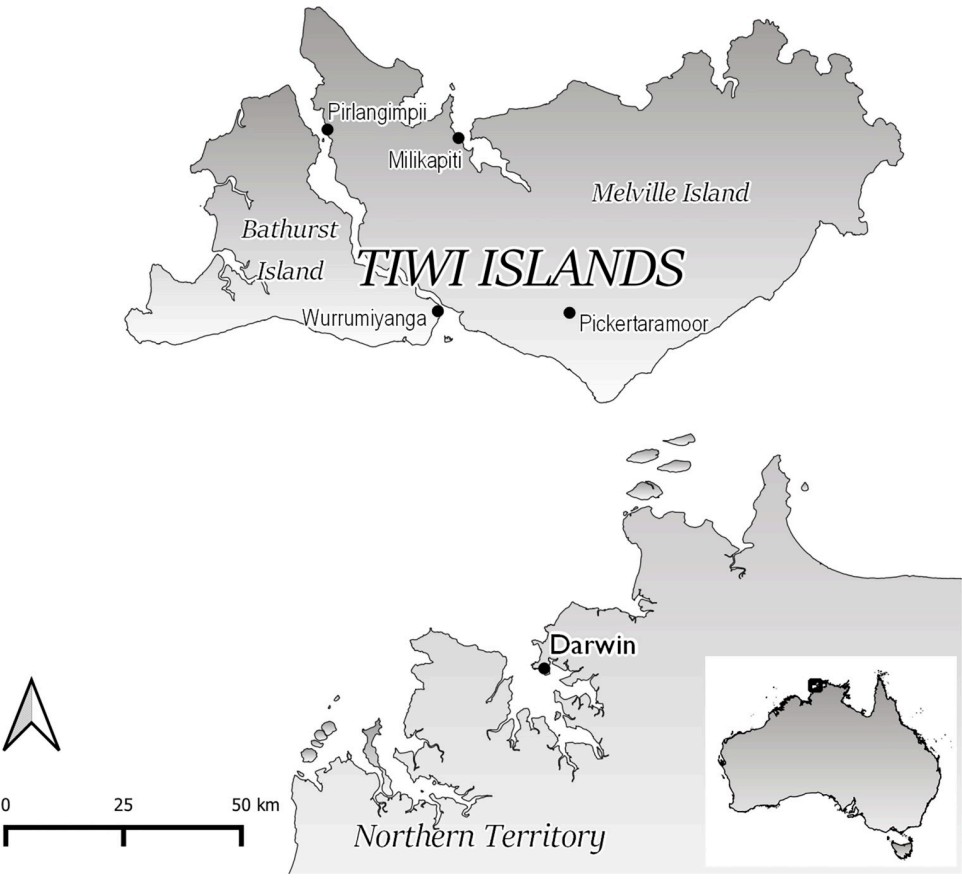

**Fig 1.** Map of the Tiwi Islands showing its proximity to mainland Australia and the Northern Territory. The Tiwi Islands comprise Bathurst Island and Melville Island. The locations of major settlements on these islands are shown. Darwin is the capital city of the Northern Territory. The map was created in QGIS version 3.26.2-Buenos Aires (QGIS Development Team (2022). QGIS Geographic Information System. Open Source Geospatial Foundation Project. http://qgis.osgeo.org). The base layer of the map used to generate this figure was downloaded from https://www.abs.gov.au/statistics/standards/australian-statistical-geography-standard-asgs-edition-3/jul2021-jun2026/access-and-downloads/digital-boundary-files and is licensed under a Creative Commons Attribution 4.0 International licence.

in 5 mL of virus transport medium [30] using a sterilised pestle. Samples were clarified by low-speed centrifugation (1000×g, 5 min, 4°C) and the supernatant used for total nucleic extraction and virus isolation.

## RNA extraction and reverse transcription quantitative PCR (RT-qPCR)

At ACDP, RNA was extracted from a sample of brain homogenate, as previously described [31]. Real-time RT-qPCR testing was performed using JEV-specific assays [32,33]. Briefly, PCR testing was performed in 96-well plates in a 25 μl reaction volume containing 5 μL of RNA, 12.5 μL of AgPath One-step RT-qPCR buffer (Ambion), 1 μL of 25X reverse transcriptase (RT), 1.0 μL of 10 μM each primer, 1.0 μL of 5 μM TaqMan probe, and 3.5 μL of nuclease free water. The following cycling conditions were used: 10 min at 45°C for reverse transcription, 10 min at 95°C for inactivation of RT, followed by 45 cycles of 95°C for 15 s, 60°C for 45 s using a 7500 Real-time PCR system (Applied Biosystems). Ct values less than 40 were considered positive.

At PWLM, RNA was extracted from each sample of brain homogenate following a column-based extraction protocol (Roche High Pure Viral Nucleic Acid Kit) with final elution volume of 60 μL. Diagnostic real-time RT-qPCR testing was performed using two JEV specific RT-qPCR assays based on the methods of [34] and [35]. The assay based on [34] was performed as a multiplex tandem PCR assay targeting the non-structural protein 5 gene (NS5) region to detect Murray Valley encephalitis virus (MVEV), Kunjin virus (KUNV), JEV and 3' untranslated region (3'UTR) to detect West Nile virus (WNV). This assay utilised a two-step PCR system consisting of a limited multiplex RT-qPCR followed by a single target real-time PCR with specific probes in the second step of the assay. The second-round individual real-time PCRs of the assay include the same primers as the first-round with the addition of hydrolysis probes for each virus. Briefly, 20 μL volume consisting of 1x reaction buffer (ThermoFisher Scientific), 10 U RNAsin (ThermoFisher Scientific), 0.3 μL One-step SS RT enzyme (ThermoFisher Scientific), 0.5 U iSTAR Taq (Scientifix Australia), 0.2 μM of each primer (MVE-F, MVE-R, WN-10533, WN-10606, KUN-F, KUN-R, JE-F, JE-R, MS2-F and MS2-R), 2.5% DMSO and 8 μL of RNA sample. Amplification was performed in a Kyratec (Kyratec) thermal cycler under the following conditions: 50°C for 30 min, 95°C for 5 min, followed by 20 cycles of 94°C for 30 s, 50°C for 30 s and 68°C for 45 s. Following amplification, the first-round PCR products were diluted 1:10 with molecular biology grade water to reduce the transfer of possible non-specific products into the second-round mixes. The second-round individual real-time TaqMan PCRs were performed in 20 μL volumes consisting of 1x PCR buffer (ThermoFisher Scientific), 4 mM MgCl$_2$, 0.2 mM dNTPs (Fisher Biotec Australia), 0.75 U DNA polymerase (ThermoFisher Scientific), 0.2 μM of forward primer (MVE-F or WN-10533 or KUN-F or JE-F, and MS2-F), 0.2 μM of reverse primer (MVE-R or WN-10606 or KUN-R or JE-R, and MS2-R), 0.2 μM of TaqMan probe (MVE-Probe or WN-10560-Probe or KUN-Probe or JE-Probe, and MS2-Probe), 0.01% BSA, and 1 μL of diluted first-round PCR product. Amplification was performed using a CFX96 Touch Real-Time PCR Detection System (Bio-Rad) thermal cycler under the following conditions: 95°C for 10 min, followed by 35 cycles of 94°C for 10 s, 55°C for 90 s and 72°C for 15 s. Fluorescence was measured at the end of the 72°C extension step. Ct values less than 35 were considered positive.

The assay based on the methods of [35] was performed as a one-step RT-qPCR in a 96-well plate format. The 20 μL reaction volume comprised 8 μL of extracted nucleic acid, 10 μL Quanta qScript XLT One-Step RT-qPCR Tough Mix (Quantabio, USA), 0.02 μL of 500 μM of each primer and 0.04 μL of 100 μM TaqMan probe. Thermocycling conditions were as follows: 10 min at 50°C, 1 min incubation at 95°C then 40 cycles of 20 s at 95°C and 80 s at 60°C using

a CFX96 Touch Real-Time PCR Detection System (Bio-Rad). All tests comprised duplicate positive control samples as well as a non-template control sample interspersed every 5 samples. Fluorescence data was automatically collected in the annealing–extension phase of the of each PCR cycle, Ct values less than 40 were considered positive.

## Virus isolation

Virus isolation from brain homogenate was attempted using both cell culture and embryonated chicken egg (ECE) culture. At ACDP, cell culture isolation was performed by inoculation of a sample of brain homogenate onto *Aedes albopictus* C6/36 cell monolayers (ATCC CRL-1660) followed by passage onto C6/36 cells and Vero cells (ATCC CCL-81) and a third and final pass from the second C6/36 passage supernatant onto Vero cells and BHK-21 cells (ATCC CCL-10). All cell lines were cultured in 25 cm$^2$ flasks, either at 37˚C (Vero, BHK-21) or 28˚C (C6/36). Vero cells were cultured in EMEM (Gibco, ThermoFisher Scientific) containing 10% foetal calf serum (FCS; Gibco, ThermoFisher Scientific), supplemented with 1% v/v L-glutamine (Sigma-Aldrich), 10 mM HEPES, 0.25% v/v penicillin–streptomycin (Sigma-Aldrich) and 0.5% v/v amphotericin B (Sigma-Aldrich). BHK-21 and C6/36 cells were cultured in BME and M199 media, respectively (Gibco, ThermoFisher Scientific), and supplemented as for Vero cells, except for C6/36 cells which also contained 1% v/v non-essential amino acids (Gibco, ThermoFisher Scientific). For virus isolation, growth media was removed from C6/36 cell monolayers followed by washing with PBS and inoculation with 200 µl of brain homogenate. Following incubation for 45 min to allow virus adsorption, inoculum was removed, and cells were washed with PBS, then overlaid with culture media containing supplements and 1% (v/v) FCS. Following 10 days incubation (pass 1), cells were frozen and thawed, and the cell suspension was centrifuged at 1000×g at 4˚C to remove debris. Clarified supernatant (1 ml) was then passaged onto a fresh cell monolayer of either C6/36 or Vero cells and incubated for 10 or 7 days, respectively (pass 2). C6/36 cells were then processed as before, and supernatant was passaged onto fresh Vero and BHK-21 cell monolayer for the third and final passage. Cells were incubated at 37˚C for 7 days and observed regularly for signs of cytopathic effect (CPE) by light microscopy. Clarified passage 3 samples were tested by real-time RT-qPCR [33] for JEV RNA detection.

The chorioallantoic membrane (CAM) of triplicate 9–11 day old specific-pathogen free ECEs were inoculated with 200 uL brain homogenate and incubated for 3 days at 37˚C. ECEs were chilled or frozen, prior to harvesting CAMs, which were visually inspected for the presence of plaques and then processed as described above for cell monolayers before the next pass. A total of 3 passages were performed and clarified supernatant from processed CAMs was tested by RT-qPCR [33] following each passage.

At PWLM, virus isolation was attempted on brain homogenate samples by inoculation of Vero cells as described above and monitored daily for CPE. Following 7 days of incubation, each culture was passed onto freshly prepared Vero cell monolayers, and this was repeated for a total of three passages. Passage three samples were extracted and tested by real-time RT-qPCR [35] to detect the presence of JEV RNA.

## Virus discovery using next generation sequencing

At ACDP, total RNA was extracted from a homogenate sample of the right thalamus of the brain using the MagMAX extraction protocol (ThermoFisher). Ribosomal RNA was depleted from the extract using the NEBNext rRNA Depletion Kit (Human/Mouse/Rat) (New England Biolabs), according to the manufacturer's instructions. A TruSeq RNA Library Prep Kit v2 (Illumina) was then used to construct libraries, which were sequenced using a P2 Reagent

Cartridge (300 cycles) on an NextSeq2000 Instrument (Illumina). Base calling and demultiplexing of the reads were performed directly on the NextSeq2000 instrument. Human sequences were removed from the raw reads by mapping to the human reference genome, build 38 (NCBI Accession GCF_000001405.39) using the *bbduk.sh* command in BBMap v.38.84 (sourceforge.net/projects/bbmap). The remaining reads were then processed by removing TruSeq adapter sequences, trimming sequence with a quality score less than 20 and finally discarding reads shorter than 50 bp with Trimmomatic v.0.38 [36]. The resultant reads were then assembled with Trinity v.2.8.5 [37] using the default parameters. Assembled transcripts were annotated using DIAMOND blastx v.0.9.24.125 [38] with the NCBI non-redundant database (accessed 2021-07-18) and a transcript matching to the JEV genome was extracted for further analysis. To ensure an accurate assembly, the cleaned reads were mapped to the assembled JEV genome using Bowtie v.2.3.4 [39] and checked for inconsistencies in Geneious v.10.2.3.

At PWLM, a previously published method was used for the preparation of metagenomic sequencing libraries [40]. Briefly, 8 μL of nucleic acid extract was DNAse treated using 1 μL ezDNase, 1 μL 10X ezDNAse buffer (ThermoFisher Scientific) at 37˚C for 10 minutes. Depletion of host and bacterial ribosomal RNA was performed by spiking 1 μL of QIAseq FastSelect probes (Qiagen) and incubating for 14 minutes in a step-down incubation from 75˚C– 25˚C, as per manufacturer's instructions. Complimentary DNA (cDNA) was reverse transcribed from treated RNA by adding 1 μL of SuperScript IV VILO Master mix (ThermoFisher Scientific), 4 μL of nuclease free water and incubating at 25˚C for 10 min, followed by 50˚C for 20 min, 85˚C for 2 min. Following cDNA synthesis, second-strand synthesis was performed by adding 8 μL Sequenase buffer, 1 μL of (1:3 ratio) Sequenase enzyme and 11 μL nuclease free water to the cDNA sample, followed by a slow 2 min ramp to 37˚C, an 8-min incubation and then a 2 min inactivation step at 95˚C. A purification step was performed on the resulting double-stranded cDNA using a 1X ratio of AMPure XP beads (Beckman Coulter), eluted in 12 μL nuclease free water. Purified ds-cDNA was quantified using a Qubit fluorometer (ThermoFisher Scientific). Libraries were constructed using the Nextera XT kit (Illumina) and sequenced on an Illumina iSeq100 platform. Base calling and demultiplexing of reads were performed automatically on the Illumina iSeq100 instrument and sequencing quality metrics were assessed using FastQC v0.11. Human sequence read removal, adaptor and quality trimming were performed as described above. The remaining reads were then *de novo* assembled using Megahit (v.1.1.3) with default parameter settings. Assembled contigs were queried against the entire NCBI non-redundant reference database (accessed 18-06-2021). Taxonomic classification of assembled contigs and individual reads were performed and visualized using MEGAN v.6.4.9. Contigs were indexed and all sequencing reads from the sample were re-mapped using minimap2 (v.2.17). The generated genome was annotated using Geneious Prime (v.2021.1) and manually inspected for accuracy.

## Tiling amplicon-based JEV genome sequencing

Complementary DNA was synthesised from sample nucleic acid extract using the SuperScript IV VILO Master Mix System (ThermoFisher Scientific) and used as input for the tiled amplicon PCR. Six microlitres of cDNA was added to each of eight singleplex PCRs, adhering to the primer scheme described in S2 Table and PCR was performed using the Platinum SuperFi II green master mix (ThermoFisher, Australia). The expected 1.5 Kb amplicon products from each PCR was verified by 1% E-gel cartridge (ThermoFisher Scientific) and then pooled. Amplicons were then purified using a 0.8X ratio of AMPure XP beads (Beckman Coulter) and quantified using the Qubit 2.0 instrument prior to library preparation using Nextera XT DNA

library preparation kit (Illumina, CA, USA). Size distribution assessment of the library was performed using a Tape Station (Agilent). The pooled library was sequenced using an iSeq 100 Reagent v2 (300 cycles) on an iSeq 100 Instrument (Illumina). Base calling and demultiplexing of reads were performed as described above and sequencing quality metrics was assessed using FastQC v0.11.1 with only high-quality libraries used in the downstream analyses. Adaptor and quality trimming were performed using the BBDuk.sh command (v38.84). Resulting paired-trimmed reads were aligned to a JEV reference genome (LC461961.1), using minimap2 (v.2.17), and mapped alignment files were indexed and sorted using SAMtools (v.1.6). Primer sequences were trimmed from the sorted read mapped alignment file using iVar (v.1.2.1). Primer sequences were identified by comparing the mapped position of the input alignment file to a reference position of primer sequences specified in a BED file. Primer-clipped alignment files were imported into Geneious Prime (v2021.1) for visual inspection prior to consensus calling. Consensus sequences were generated in Geneious Prime with parameters set at 10 for minimum depth, 20 for minimum quality, and 25% for minimum frequency.

The final consensus whole genome sequence was deposited in the NCBI GenBank database (Accession no. OM867669).

## Phylogenetic analysis

The complete genome of JEV/Australia/NT_Tiwi Islands/2021 (10,949 nt) was aligned with all complete JEV genomes available on NCBI's GenBank using MAFFT v.7.301 [41], with the auto flag to select the optimal alignment parameters. Maximum likelihood phylogenetic trees were created using IQ-TREE v.2.0.6 [42] with 1,000 bootstrap replicates and an evolutionary model selected by IQ-TREE. The phylogenetic tree was then visualised and rendered with the R package ggtree v.1.14.6 [43].

An expanded phylogenetic tree was generated using complete envelope (E) gene (1,500 nt) sequences, available in NCBI GenBank. The tree was generated as described above, except that the transition model 2 (TIM2) with gamma rate heterogeneity was chosen as the most appropriate evolutionary model by IQ-TREE.

A maximum clade credibility (MCC) phylogeny was re-constructed from a dataset of 519 spatio-temporally defined complete E gene sequences. With the removal of temporal outlier Genotype V (GV) sequences, the resultant dataset demonstrated sufficient temporal signal for further Bayesian Markov chain Monte Carlo (MCMC) analysis, as defined in temporal regression analysis within TempEst v.1.5.3. MCMC analysis was conducted using the BEAST v.1.10 package. An uncorrelated relaxed molecular clock was employed assuming a GTR+G+I nucleotide substitution model and a constant prior. Three independent chains of $5 \times 10^8$ generations were assembled, and subsequently assessed for convergence. A 10% burn in was used when reconstructing the MCC phylogeny. The resultant MCC phylogeny was illustrated using FigTree v1.4.3.

## Selection analysis

A total of 361 whole genomes (including JEV/Australia/NT_Tiwi Islands/2021) were analysed for evolutionary selection using the HyPhy package through the DataMonkey webserver (http://www.datamonkey.org/). Several methods were used to test for positive (diversifying) and episodic selection, including Single-Likelihood Ancestor Counting (SLAC), Fixed Effects Likelihood (FEL), Fast Unconstrained Bayesian AppRoximation (FUBAR) and the Mixed Effects Model of Evolution (MEME). SLAC and FEL use combinations of counting and Maximum Likelihood (ML) approaches to infer non-synonymous (dN) and synonymous (dS) substitution rates and assume that selection pressure is constant [44]. FUBAR employs a

Bayesian approach to infer dN and dS substitution rates, while MEME uses a mixed-effects ML approach to test for episodic positive or diversifying selection [45,46].

## Results

### Diagnostic virology

Samples from different locations of the brain were tested using a range of JEV- or flavivirus-specific RT-qPCR tests at ACDP and PWLM. Of the five assays employed, only the Shao et al. (2018) [33] and Bharucha et al. (2018) [35] assays detected JEV in the samples tested (Table 1). Only the right thalamus sample was available for testing using the assay reported by Shao et al. [33]. Virus isolation attempts using mammalian cells (Vero) and mosquito cells (C6/36) followed by passage in mammalian cell lines (Vero, BHK-21) were unsuccessful. We also attempted isolation from the right thalamus specimen via inoculation of the CAMs of ECEs, a culture system that has previously been shown to be more sensitive than cell culture and suckling mouse inoculation methods for isolating MVEV, a closely related flavivirus to JEV, from human brain specimens [47,48]. Unfortunately, virus was also unable to be isolated using this ECE culture system.

### RNA-seq and amplicon-based sequencing

At ACDP, whole genome sequencing of the JEV genome was performed using nucleic extract from the RT-qPCR-positive thalamus sample. An untargeted NGS approach was used to generate 16,613,558 reads following quality trimming and filtering. Most of these reads were non-viral (99.92%); however, the sequencing depth achieved allowed the assembly of a complete 10,949 nt JEV genome, with an average sequencing depth of 197x. At PWLM, the midbrain sample, returning the lowest Ct value, was selected for unbiased RNA sequencing. The sample library generated 556,558 reads following quality trimming and filtering. While 99.96% of the homogenate sample were non-viral reads, 0.03% reads mapped to 66% of a JEV reference genome at an average sequencing depth of 2.5x (range:2-9x). To enrich the JEV genome and improve the likelihood of generating a complete JEV genome sequence, a custom designed 1.5kb tiled amplicon workflow was employed. This approach generated a total of 232,708 high quality reads mapping to 99.96% of a JEV reference genome (LC461961) at an average read depth of 3211x. Pairwise sequence alignment revealed that genomes derived from ACDP and PWLM were identical.

### Phylogenetic and evolutionary analyses

A phylogenetic reconstruction of the full coding sequence of JEV was carried out to determine the genotype of the Tiwi Islands strain (designated JEV/Australia/NT_Tiwi Islands/2021) and

**Table 1. Results of JEV RT-qPCR testing of brain samples.**

| Sample | RT-qPCR assay results (Ct values)[a] | | | |
|---|---|---|---|---|
| | Shao et al., 2018 [33] | Pyke et al., 2004 [32] | Harnett and Cattell, 2010 [34] | Bharucha et al. 2018 [35] |
| Medulla | NT | NT | Neg | 28.4 |
| Pons | NT | NT | Neg | 29.1 |
| Right thalamus | 24.4 | Neg | Neg | 28.7 |
| Midbrain | NT | NT | Neg | 28.2 |
| Right cerebellum | NT | NT | Neg | 32.3 |

[a]Refer to the Methods section for details of each assay used

[b]NT, not tested

its relatedness to previously reported strains belonging to JEV genotypes I to V (n = 369). This analysis revealed that the JEV/Australia/NT_Tiwi Islands/2021 genome sequence was a divergent member of JEV GIV with a basal phylogenetic position relative to the closest related sequences (Fig 2). BLASTn analysis revealed that the Tiwi Islands genome was most closely related (99.8% nucleotide identity) to a recent genome sequence derived from an aborted piglet at an Australian piggery in 2022 (ON624132). The Tiwi Islands genome shared next highest levels of nucleotide identity with JEV strains of Indonesian origins derived from porcine (LC461961; 96.8%) and human infections (MT253731; 96.7%) acquired in Bali in 2017 and 2019 [49], respectively.

To maximise spatiotemporal sampling of the JEV GIV viruses, this analysis was expanded to include all publicly available JEV E gene sequences (n = 1181), which are more numerous than published whole JEV genomes. This confirmed the placement of the Tiwi Islands strain

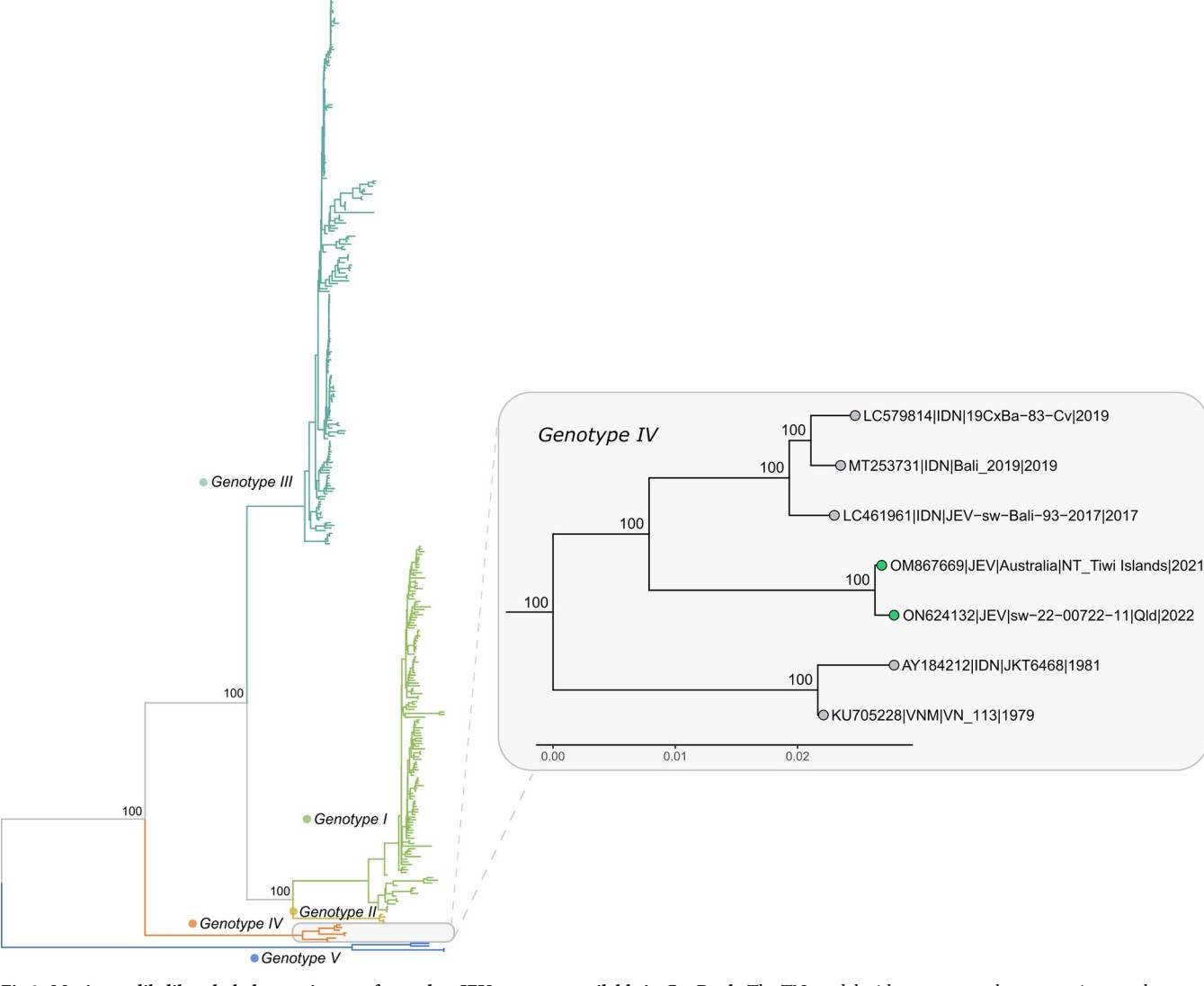

**Fig 2. Maximum likelihood phylogenetic tree of complete JEV genomes available in GenBank.** The TN model with gamma rate heterogeneity was chosen as the most appropriate model by IQ-TREE v.2.0.6. The results from 1000 bootstrap replicates are given on the nodes and the scale represents the number of nucleotide substitutions per site.

within GIV as a divergent sub-lineage, sharing a common ancestral sequence with Balinese isolates from 2017 and 2019 (S1 Fig).

A maximum clade credibility (MCC) phylogeny was reconstructed from a dataset of 519 spatio-temporally defined complete E gene sequences (Fig 3). Sequences clustering within genotype 5GV were excluded from this analysis, as they were outliers in temporal regression analysis of the initial JEV dataset. The evolutionary rate of the clade containing the JEV Tiwi Islands sequence was estimated as $9.95 \times 10^{-4}$ substitutions/site/year (95% highest probability density (HPD): $5.31–14.9 \times 10^{-4}$), with the most recent common ancestor (MRCA) emerging approximately in the year 2007 (mean 2007, 95% HPD 1998–2014), or around 13 years prior to the fatal case of JE on the Tiwi Islands.

## Positive selection in the E and NS proteins

The ratio of non-synonymous (dN) to synonymous (dS) nucleotide substitutions (dN/dS) was estimated to be 0.0635 in the SLAC DataMonkey analysis, pointing toward predominately negative (purifying) selection, as observed for other flaviviruses [50–52]. This was also seen in the SLAC and FUBAR analyses, which both detected sites under negative selection but no sites under positive (diversifying) selection. MEME and FEL detected positive selection at several sites in the pre-membrane (prM), E, NS1, NS2a, NS3, NS4a and NS5 protein genes (S3 Table). Notably, the S194N site of the E protein is unique to the Australian GIV viruses and is located in the hinge region between domains I and II of the three-dimensional structure [53].

## Discussion

The finding that the JEV from the Tiwi Islands belonged to genotype IV was unexpected. Phylogenetic analysis of the Tiwi Islands JEV genome revealed it to be a highly divergent strain of JEV GIV, occupying a distinct lineage within this genotype, together with the 2022 Australian outbreak strain (Figs 2, 3 and S1). Notably, the Tiwi Islands strain was found to have the next closest phylogenetic relationship with a group of viruses originating from Bali, Indonesia, comprising viruses derived from a 2019 human JE case [49], a pool of *Cx. vishnui* collected in 2019 [54], and from pig sera collected in 2017 [55]. Together with the known distribution of genotype IV viruses, these findings support the ancestral origin of the Australian strain as Indonesia. In our Bayesian analysis, the Tiwi Islands sequence occupied a basal position within a clade that contained the 2017 and 2019 Indonesian JEV sequences (Fig 3). The estimated emergence of this clade was 2007 (95% HPD: 1998–2014), suggesting that this lineage of GIV viruses has been circulating for several years before detections in Bali and the Tiwi Islands. The fine scale resolution required to more accurately estimate the time of emergence of the ancestral strain of the Tiwi Islands virus is hindered by the scarcity of JEV GIV sequence data. Regional surveillance for JEV will also be pivotal in understanding the origins of the Tiwi Islands strain, and the extent of genetic diversity and geographic spread of JEV GIV. To date, our knowledge of JEV activity in the countries neighbouring Australia's northern borders has been informed by case reports or through sporadic or opportunistic sampling and testing as part of surveillance programs or discrete research studies [17–19,21]. Thus, there are no published reports of the detection of JEV GIV from Timor-Leste, West Papua, or Papua New Guinea, all of which lie to the north of Australia.

This is the third genotype of JEV to have been reported as acquired from virus circulating in Australia. Previous incursions of JEV into the Torres Strait and Cape York of northern Queensland had either been members of GII in 1995 and 1998 [12,15] or GI in 2000 and 2004 [23,56]. The Tiwi Islands case was the first definitive occurrence of the GIV lineage outside Indonesia and only the second fatal case reported by a GIV virus; the first was a fatal case in a

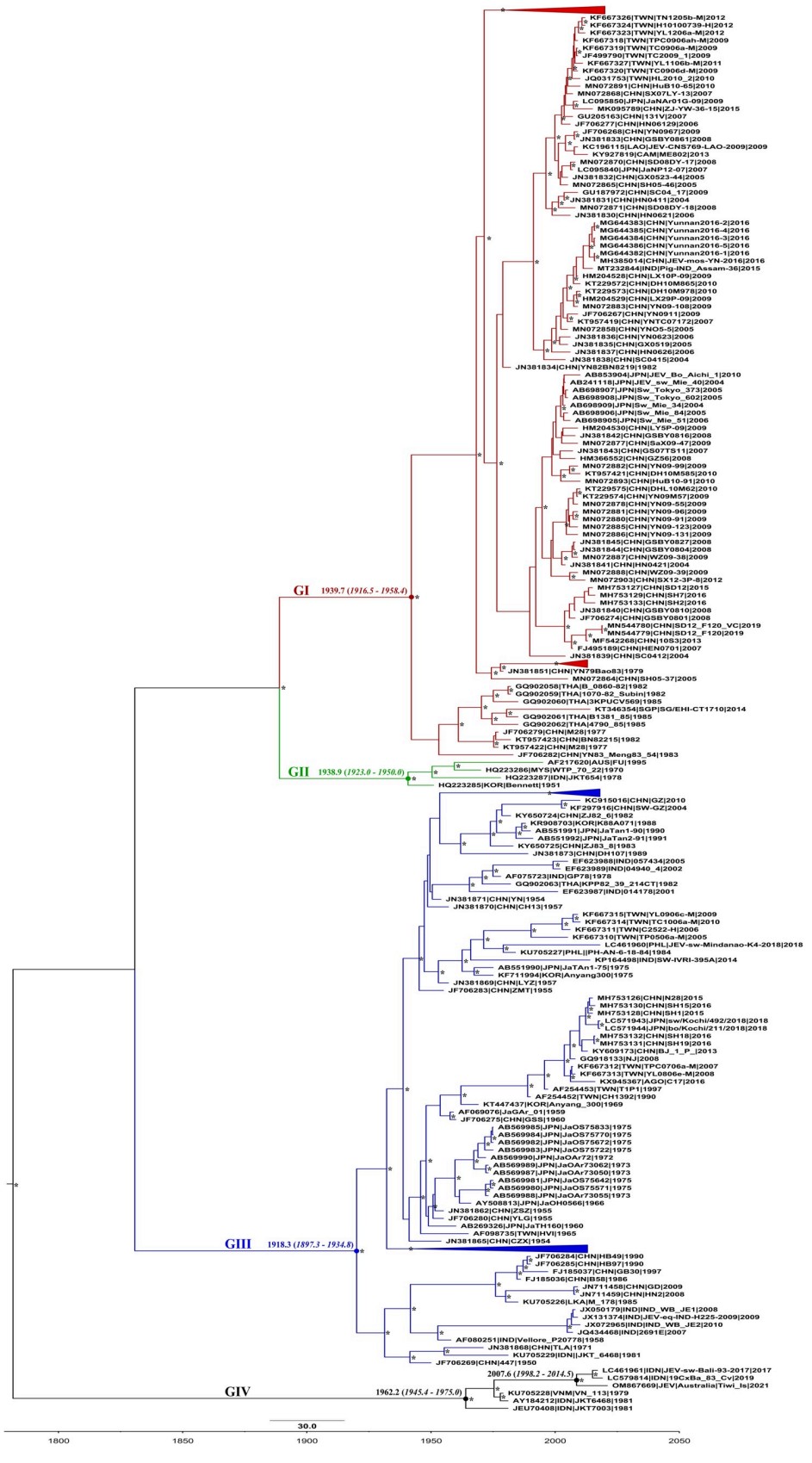

**Fig 3. Maximum clade credibility phylogenetic tree of complete JEV envelope gene sequences, sourced from GenBank.** The phylogeny was reconstructed under a GTR+G+I nucleotide substitution assumption and an uncorrelated relaxed molecular clock model. Genotype V sequences of JEV were excluded from this phylogeny as they were deemed temporal outliers in regression analysis. Posterior probability values of >0.70 are presented adjacent to nodes as indicated by asterisks. The mean time to most recent common ancestor (tMRCA) is presented above major nodes, with error reported as the 95% highest probability density (95% HPD).

tourist who had visited Bali in 2019 and was subsequently diagnosed after returning to Queensland, Australia [49]. A third human case due to a GIV virus is believed to have occurred in Hanoi, in 1979, but the details and outcome of this case are unknown (personal communication from Dr K Plante; the virus isolate had been provided to the Arbovirus Library at the University of Texas Medical Branch by Dr J Landinshy in 1979). All other known GIV isolates had been obtained from *Cx tritaeniorhynchus*, *Cx. vishnui* or unidentified mixed pools of mosquitoes trapped in Java, Bali or Flores, between 1980 and 1981, and more recently from pigs in 2017 and *Cx. vishnui* in 2019 [54,55].

It is not yet clear whether the emergent GIV strain has displaced the previously circulating genotypes in Australia. Such a phenomenon has been observed for the emergence of GI as the dominant genotype in Southeast Asia in the mid-1990s [7,9]. The biological mechanism associated with GI dominance has not been clearly established, but may involve enhanced replicative fitness in mosquito vectors, based on studies of *in vitro* replication kinetics [9,57]. Since no isolate has been available, the biological properties of the emergent Australian GIV strain have not yet been able to be investigated, but future studies may reveal viral or host factors that underly a fitness advantage of this GIV strain over pre-existing genotypes circulating in the Australasian region. To investigate a genetic basis to the emergence of the Tiwi Islands JEV strain, we performed selection analyses that detected positive selection using two methods (MEME and FEL) at several sites in the gene sequences of prM, E and NS proteins (S3 Table). Of potential importance is the single positively selected site in the E protein (S194N), which is unique to the Australian GIV strains and encodes a non-conservative amino acid substitution located in hinge 3 linking domains I and II of the E protein [53]. The DI-DII hinge plays an important structural role during membrane fusion of the flavivirus particle and contains neutralising epitopes [53,58] and may prove to contain important molecular determinants of replicative fitness and virulence for vectors or vertebrate hosts.

It is not known how JEV GIV was introduced into the Tiwi Islands of northern Australia in 2021. There are several possible mechanisms by which the virus could have reached and potentially established in the Islands, including by migratory or vagrant ardeid birds, by the transport of infected mosquito vectors by air or ship, or by infected mosquito vectors being blown by cyclonic winds from endemic areas in Indonesia. Ardeid birds, especially herons and egrets, are major reservoir hosts of JEV. The Nankeen night heron (*Nycticorax caledonicus*) is one such species that may play an important role in the introduction and spread of JEV into new areas. Populations of this species may be partially migratory or nomadic with a range extending from Australia to Papua New Guinea and parts of Indonesia [59,60], and have been shown to develop viraemia following experimental infection [61]. Infected mosquitoes are also known to be transported occasionally by aircraft, and this may have been the means by which JEV reached Angola [62] and Saipan [63] and Italy [64,65]. No regular movement of aircraft are known to occur between the Tiwi Islands and either Indonesia or Papua New Guinea, making this mechanism unlikely. Similarly, commercial international shipping into the Tiwi Islands no longer occurs. However, introduction via unregulated or illegal shipping or private boats cannot be discounted.

The most likely source of wind-blown infected mosquitoes into the Tiwi Islands is Indonesia or Timor Leste, although Papua New Guinea cannot be ruled out as a potential source. JEV

is believed to be enzootic in Indonesia, with virus isolations from mosquitoes or serological evidence of infection in pigs found in parts of Java, Bali, Lombok, Flores, West Timor and Papua Province (reviewed in [10,11,66]). A clinical case of JE from Irian Jaya (West Papua) was reported from Timika in 1996 [21], supported by serological evidence of human infection [67]. JEV has also been shown to be relatively widespread in Papua New Guinea since the mid- to late-1980s from serological investigations [18], and from virus isolations and human cases reported from Western Province [17] and Port Moresby, National Capital District [10,19,20]. Tropical depressions and cyclonic events were suggested to have been the probable route of introduction of JEV from Papua New Guinea into the Torres Strait in 1995, 1998 and 2000 [12,15,23] and western Cape York Peninsula in 1998 [22]. Modelling of the possible movement suggested that mosquitoes could be carried in monsoonal winds at a height of 300 m for more than 650 km between Papua New Guinea and the areas of the western Cape York Peninsula. In support of this, Kay and Farrow reported circumstantial evidence that *Cx. annulirostris* mosquitoes, the main vector of JEV in the Torres Strait and Australia, had a flight range of between 594–648 km under suitable conditions based on aerial collections [68]. Other studies had also reported migratory *Cx. tritaeniorhynchus*, the major vector of JEV, carried by monsoonal winds over significant distances [69,70]. The recent introduction of *Culex tritaeniorhynchus* mosquitoes into the Northern Territory was believed to have been carried by wind from Timor Leste, approximately 620km north of Australia [71], although harbourage in aeroplanes and importation into Darwin from Bali or other northern locations is also possible [72].

In response to the Tiwi Islands case of JE, laboratory testing of samples collected from feral pigs and buffalo on the Tiwi Islands was undertaken by the Northern Australian Quarantine Strategy (NAQS) program in June 2021. No evidence of specific JEV infection was found from RT-qPCR and serological testing of these samples. However, sentinel cattle located at Beatrice Hill, near Darwin, that were sampled between November 2020 and April 2021 were positive for JEV-specific antibodies, as well as a single feral pig sampled from Croker Island, approximately 100 km east of the Tiwi Islands, as part of a routine NAQS survey in November 2020 (Dr S. Fruean [NAQS] and Dr. V. Bhardwaj [Berrimah Veterinary Laboratory], personal communication). Thus, although evidence of prior or active infection with JEV in animals sampled from Tiwi Islands was not found in the months after the human case, the limited serological evidence in animals at other locations in the Northern Territory suggests low level transmission of JEV during the 2020/21 wet season.

Initial rounds of molecular testing of post-mortem brain homogenate samples at PWLM and ACDP revealed that two of the RT-qPCR assays in routine use failed to detect the novel Tiwi Islands strain of JEV (Table 1). *In silico* analysis demonstrated multiple target mismatches in each of the forward, reverse and probe binding regions of these assays (S2 Fig). The findings from this study emphasise the utility of unbiased metagenomic next generation sequencing for detecting and characterising genetically distinct viruses that may be missed by PCR-based approaches. In addition, metagenomics can serve as an important tool to help guide the design of new molecular tests and confirm the specificity of existing tests. In this study the metagenomic sequencing approach facilitated the design of the amplicon-based sequence methodology and informed the choice of RT-qPCR screening assay. These findings also highlight the importance of ongoing *in silico* assessment of molecular assays, especially for pathogens that may not be present locally but are circulating in neighbouring regions.

The Tiwi Islands case of JE in 2021 and the subsequent large multi-state outbreak in 2022 highlights the risk of incursions of vector-borne diseases into Australia's northern regions and underscores the importance of ongoing surveillance for both endemic and exotic arboviruses and the need to ensure that state and territory public and animal health laboratories have the capability to identify emerging viruses. Further knowledge is urgently needed to better

understand the ecology of JEV and other flaviviruses in northern Australia and the pathways by which they are introduced and subsequently spread. In particular, there are major knowledge gaps for the range of avian species that play a role as maintenance hosts and in spreading the virus over long distances, the role of other species in transmission cycles, such as megachiropteran and microchiropteran bats, and the identification of competent vector species. With respect to the latter, two mosquito species that are well-established as major vectors of JEV in Asia, *Cx. tritaeniorhynchus* and *Cx. gelidus*, have both been found as exotic mosquitoes that have entered and become established in northern Australia [24,71], but there is little information on their incidence and geographic occurrence. Other than *Cx. annulirostris*, few studies have been undertaken on the competence of other mosquito species to transmit the different JEV genotypes [73].

## Supporting information

**S1 Fig. Maximum likelihood phylogenetic tree of JEV envelope genes available in Gen-Bank.** The TIM2 model with gamma rate heterogeneity was chosen as the most appropriate model by IQ-TREE v.2.0.6. The results from 1000 bootstrap replicates are given on the nodes and the scale represents the number of nucleotide substitutions per site.
(TIF)

**S2 Fig. Sequence alignment of primers and probes used for the detection of JEV at PWLM and ACDP.** Primer and probe positions are shown according to the sequence of the JEV/Australia/NT_Tiwi Islands/2021 genome (OM867669). JEV RT-PCR assays used at PWLM comprised primers and probes in alignments 1 and 2 [34,35]. JEV RT-PCR assays used at ACDP comprised primers and probes in alignments 3 and 4 [32,33].
(TIF)

**S1 Table. Japanese encephalitis virus detections in Australia.**
(DOCX)

**S2 Table. Primer scheme used for the amplification of JEV 1600bp amplicon sequencing.**
(DOCX)

**S3 Table. Positively selected sites within the JEV/Australia/NT_Tiwi Islands/2021 genome.**
(DOCX)

## Acknowledgments

The authors wish to thank Dr Marianne Tiemensma of the Forensic Department, Royal Darwin Hospital for pathology support and Dr Claire Waller of the Infectious Diseases Department, Royal Darwin Hospital for clinical support; the Diagnostic Virology and Molecular Diagnostic teams at the Australian Centre for Disease Preparedness for laboratory technical support; the Molecular Diagnostic team at PathWest Laboratory Medicine WA for laboratory technical assistance; and Dr Peter Durr and Dr Kerryne Graham (ACDP) for the map shown in Fig 1. The authors acknowledge the capabilities of the Australian Centre for Disease Preparedness (grid.413322.5) in undertaking this research, including infrastructure funded by the National Collaborative Research Infrastructure Strategy.

## Author Contributions

**Conceptualization:** Chisha Sikazwe, Matthew J. Neave, Avram Levy, Allison Imrie, Robert W. Baird, Bart J. Currie, David Speers, John S. Mackenzie, David W. Smith, David T. Williams.

**Data curation:** Chisha Sikazwe, Matthew J. Neave, Patrick Mileto.

**Formal analysis:** Chisha Sikazwe, Matthew J. Neave, Alice Michie, Avram Levy, John S. Mackenzie, David W. Smith, David T. Williams.

**Investigation:** Chisha Sikazwe, Matthew J. Neave, Alice Michie, Patrick Mileto, Jianning Wang, Natalie Cooper, Avram Levy, Allison Imrie, Robert W. Baird, Bart J. Currie, David Speers, John S. Mackenzie, David W. Smith, David T. Williams.

**Methodology:** Chisha Sikazwe, Matthew J. Neave, Alice Michie, Patrick Mileto, Jianning Wang, Natalie Cooper, Avram Levy, Robert W. Baird, Bart J. Currie, David Speers, David T. Williams.

**Resources:** Chisha Sikazwe, Matthew J. Neave, Avram Levy, Allison Imrie, Robert W. Baird, Bart J. Currie, David Speers, David W. Smith, David T. Williams.

**Supervision:** Chisha Sikazwe, Matthew J. Neave, Avram Levy, Allison Imrie, David Speers, John S. Mackenzie, David W. Smith, David T. Williams.

**Visualization:** Chisha Sikazwe, Matthew J. Neave, Alice Michie.

**Writing – original draft:** Chisha Sikazwe, Matthew J. Neave, Alice Michie, Avram Levy, John S. Mackenzie, David W. Smith, David T. Williams.

**Writing – review & editing:** Chisha Sikazwe, Matthew J. Neave, Alice Michie, Patrick Mileto, Jianning Wang, Natalie Cooper, Avram Levy, Allison Imrie, Robert W. Baird, Bart J. Currie, David Speers, John S. Mackenzie, David W. Smith, David T. Williams.

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
