## [Decision Letter · Decision Letter 0]

25 Sep 2022

Dear Dr. Williams,

Thank you very much for submitting your manuscript "Molecular detection and characterisation of the first Japanese encephalitis virus belonging to genotype IV in Australia" for consideration at PLOS Neglected Tropical Diseases. As with all papers reviewed by the journal, your manuscript was reviewed by members of the editorial board and by several independent reviewers. The reviewers appreciated the attention to an important topic. Based on the reviews, we are likely to accept this manuscript for publication, providing that you modify the manuscript according to the review recommendations. 

Sincerely,

Geraldine Marie Foster, PhD

Academic Editor

Elvina Viennet

Section Editor

Reviewer's Responses to Questions

**Key Review Criteria Required for Acceptance?**

**Methods**

-Are the objectives of the study clearly articulated with a clear testable hypothesis stated?

-Is the study design appropriate to address the stated objectives?

-Is the population clearly described and appropriate for the hypothesis being tested?

-Is the sample size sufficient to ensure adequate power to address the hypothesis being tested?

-Were correct statistical analysis used to support conclusions?

-Are there concerns about ethical or regulatory requirements being met?

Reviewer #1: (No Response)

Reviewer #2: The objective is clearly stated. No concerns.

Reviewer #3: -Are the objectives of the study clearly articulated with a clear testable hypothesis stated? Yes

-Is the study design appropriate to address the stated objectives? Yes

-Is the population clearly described and appropriate for the hypothesis being tested? Yes

-Is the sample size sufficient to ensure adequate power to address the hypothesis being tested? Yes

-Were correct statistical analysis used to support conclusions? Yes

-Are there concerns about ethical or regulatory requirements being met? No

**Results**

-Does the analysis presented match the analysis plan?

-Are the results clearly and completely presented?

-Are the figures (Tables, Images) of sufficient quality for clarity?

Reviewer #1: (No Response)

Reviewer #2: The results are clear and supported.

Reviewer #3: -Does the analysis presented match the analysis plan? Yes

-Are the results clearly and completely presented? Yes

-Are the figures (Tables, Images) of sufficient quality for clarity? Yes

**Conclusions**

-Are the conclusions supported by the data presented?

-Are the limitations of analysis clearly described?

-Do the authors discuss how these data can be helpful to advance our understanding of the topic under study?

-Is public health relevance addressed?

Reviewer #1: (No Response)

Reviewer #2: The conclusion is good and the discussion explains the public health relevance.

Reviewer #3: -Are the conclusions supported by the data presented? Yes

-Are the limitations of analysis clearly described? NA

-Do the authors discuss how these data can be helpful to advance our understanding of the topic under study? Yes

-Is public health relevance addressed? Yes

**Editorial and Data Presentation Modifications?**

Reviewer #1: (No Response)

Reviewer #2: I did not see any edits.

Reviewer #3: Lines 132-136 and 228-233: Prior to reviewing this manuscript, I was not familiar with the utility of using embryonated chicken eggs to isolate avian-associated flaviviruses. Please consider adding some references describing this procedure and its relative success compared to other methods of isolation (i.e., cell culture and intracranial inoculation of suckling mice). 

Lines 346-349 and Table 1: It is interesting that the Shao et al., 2018 RT-PCR assay detected JEV RNA in the right thalmus tissue sample but in no other tissue sample types. Do you happen to have an explanation for this finding (e.g., sequence mismatch in the primer-binding site)? 

Lines 418-420: Has regular surveillance for JEV in mosquitoes, birds, and/or pigs been conducted in Timor-Leste, West Papua, and Papua New Guinea? After reading this sentence, I am unclear if regular surveillance for JEV is conducted, and the virus has not been detected or if regular surveillance has not been conducted. Please clarify. 

Figures 2 and 3, and Supplementary figure 1: Please include bootstrap support values for the genotype-defining nodes.

**Summary and General Comments**

Reviewer #1: (No Response)

Reviewer #2: This paper looks ready to be published and contains interesting data.

Reviewer #3: In the manuscript “Molecular detection and and characterisation of the first Japanese encephalitis virus

belonging to genotype IV in Australia” (PNTD-D-22-01068), Sikazwe et al. reports the detection and whole genome sequencing of Japanese encephalitis virus (JEV) RNA recovered from a fatal case of JE that occurred in a resident of the Tiwi Islands, in the Northern Territory of Australia in 2021. The authors then perform an evolutionary analysis to examine the link between this new genotype IV sequence (first detection of genotype IV of JEV in Australia), other genotype IV sequences from Vietnam and Indonesia detected from 1979─2019, and a genotype IV sequence recovered from a large JE outbreak in southeastern Australia in 2022. The close phylogenetic relationship between the Tiwi Island sequence and recent Indonesian sequences suggest that Indonesia might be the origin of genotype IV viruses currently circulating in Australia. 

This manuscript is very well-written, and the study was well-executed and scientifically sound. The introduction includes all relevant information necessary to understand the study, the methods are comprehensive, the results are well-organized, and the discussion nicely discusses possible origins of genotype IV sequences currently circulating in Australia, and the need for JEV vector competence studies and ongoing surveillance for the virus. Listed below are a few very minor comments. 

Specific comments

Lines 132-136 and 228-233: Prior to reviewing this manuscript, I was not familiar with the utility of using embryonated chicken eggs to isolate avian-associated flaviviruses. Please consider adding some references describing this procedure and its relative success compared to other methods of isolation (i.e., cell culture and intracranial inoculation of suckling mice). 

Lines 346-349 and Table 1: It is interesting that the Shao et al., 2018 RT-PCR assay detected JEV RNA in the right thalmus tissue sample but in no other tissue sample types. Do you happen to have an explanation for this finding (e.g., sequence mismatch in the primer-binding site)? 

Lines 418-420: Has regular surveillance for JEV in mosquitoes, birds, and/or pigs been conducted in Timor-Leste, West Papua, and Papua New Guinea? After reading this sentence, I am unclear if regular surveillance for JEV is conducted, and the virus has not been detected or if regular surveillance has not been conducted. Please clarify. 

Figures 2 and 3, and Supplementary figure 1: Please include bootstrap support values for the genotype-defining nodes.

PLOS authors have the option to publish the peer review history of their article (what does this mean?). If published, this will include your full peer review and any attached files.

Reviewer #1: No

Reviewer #2: No

Reviewer #3: No

Figure Files:

Data Requirements:

Reproducibility:

References

---

## [Editor Report · Decision Letter 1]

24 Oct 2022

Dear Dr. Williams,

We are pleased to inform you that your manuscript 'Molecular detection and characterisation of the first Japanese encephalitis virus belonging to genotype IV acquired in Australia' has been provisionally accepted for publication in PLOS Neglected Tropical Diseases.

Best regards,

Geraldine Marie Foster, PhD

Academic Editor

Elvina Viennet

Section Editor

---

## [Editor Report · Acceptance letter]

10 Nov 2022

Dear Dr. Williams,

We are delighted to inform you that your manuscript, "Molecular detection and characterisation of the first Japanese encephalitis virus belonging to genotype IV acquired in Australia," has been formally accepted for publication in PLOS Neglected Tropical Diseases.

Best regards,

Shaden Kamhawi

co-Editor-in-Chief

Paul Brindley

co-Editor-in-Chief
